# Semaphorin4A-Plexin D1 Axis Induces Th2 and Th17 While Represses Th1 Skewing in an Autocrine Manner

**DOI:** 10.3390/ijms21186965

**Published:** 2020-09-22

**Authors:** Tiago Carvalheiro, Carlos Rafael-Vidal, Beatriz Malvar-Fernandez, Ana P. Lopes, Jose M. Pego-Reigosa, Timothy R. D. J. Radstake, Samuel Garcia

**Affiliations:** 1Department of Rheumatology & Clinical Immunology, University Medical Center Utrecht, University of Utrecht, 3508 GA Utrecht, The Netherlands; t.ferreiracarvalheiro@umcutrecht.nl (T.C.); beatrizmalvar@gmail.com (B.M.-F.); A.P.PinheiroLopes-3@umcutrecht.nl (A.P.L.); tradstake73@gmail.com (T.R.D.J.R.); 2Center for Translational Immunology, University Medical Center Utrecht, University of Utrecht, 3508 GA Utrecht, The Netherlands; 3Rheumatology & Immuno-mediated Diseases Research Group (IRIDIS), Galicia Sur Health Research Institute (IIS Galicia Sur), SERGAS-UVIGO, 36312 Vigo, Spain; carlos.rafael@iisgaliciasur.es (C.R.-V.); jose.maria.pego.reigosa@sergas.es (J.M.P.-R.); 4Rheumatology Department, University Hospital Complex of Vigo, 36312 Vigo, Spain

**Keywords:** semaphorin4A, plexinD1, CD4^+^ T cell differentiation, T helper cells, systemic sclerosis

## Abstract

Semaphorin (Sema)4A is a transmembrane glycoprotein that is elevated in several autoimmune diseases such as systemic sclerosis, rheumatoid arthritis and multiple sclerosis. Sema4A has a key role in the regulation of Thelper Th1 and Th2 differentiation and we recently demonstrated that CD4^+^ T cell activation induces the expression of Sema4A. However, the autocrine role of Sema4A on Th cell differentiation remains unknown. Naïve Th cells from healthy controls were cell sorted and differentiated into Th1, Th2 and Th17 in the presence or absence of a neutralizing antibody against the Sema4A receptor PlexinD1. Gene expression was determined by quantitative PCR and protein expression by ELISA and flow cytometry. We found that the expression of Sema4A is induced during Th1, Th2 and Th17 differentiation. PlexinD1 neutralization induced the differentiation of Th1 cells, while reduced the Th2 and Th17 skewing. These effects were associated with an upregulation of the transcription factor T-bet by Th1 cells, and to downregulation of GATA3 and RORγt in Th2 cells and Th17 cells, respectively. Finally, PlexinD1 neutralization regulates the systemic sclerosis patients serum-induced cytokine production by CD4^+^ T cells. Therefore, the autocrine Sema4A-PlexinD1 signaling acts as a negative regulator of Th1 skewing but is a key mediator on Th2 and Th17 differentiation, suggesting that dysregulation of this axis might be implicated in the pathogenesis of CD4^+^ T cell-mediated diseases.

## 1. Introduction

T helper (Th) cells are essential regulators of the adaptive immune responses. There are three major subsets of Th cells, which can be identified based on their cytokine production and immune function. Th1 cells are characterized by the secretion of interferon-γ (IFN-γ) and are involved in the host defense against intracellular pathogens; Th2 cells usually secret interleukin (IL)-4, IL-5 and IL-13 and are critical for mediating immune responses against extracellular parasites; finally Th17 cells mainly produce IL-17, IL-21 and IL-22, and are pivotal in the defense against extracellular bacteria and fungi [1,2,3]. Dysregulation of Th homeostasis has been implicated in several diseases. Hyperactivation of Th1 and Th17 cells confers susceptibility to autoimmune diseases and is associated with the pathology of multiple sclerosis (MS), inflammatory bowel disease, systemic sclerosis (SSc), psoriasis, rheumatoid arthritis (RA) and spondiloarthropaties [4,5,6,7]. On the other hand, Th2 cells are implicated in the pathogenesis of asthmatic and allergic diseases [8,9].

Semaphorin (Sema) 4A is a transmembrane glycoprotein belonging to the semaphorin family, which is a large group of proteins initially described as axonal guidance molecules [10]. Sema4A is mainly produced by dendritic cells, monocytes and macrophages [11], although it can be also expressed by activated CD4^+^ T cells and germinal center B cells [12,13,14]. Sema4A binds to different receptors expressed on CD4^+^ T cells, such as PlexinB1, PlexinB2, PlexinD1, Neuropilin-1 (NRP-1), Immunoglobulin-like transcript 4 (ILT)4 and Tim-2, although the last has been reported in mice and has no human orthologous [12,13,14,15,16,17,18]. Several studies have shown that Sema4A expression is elevated in diseases in which Th cells play an essential role, including asthma, RA, MS and SSc [11,12,13,19]. Furthermore, functional studies have demonstrated that Sema4A is a key molecule in the regulation of naïve CD4^+^ T cell differentiation, although different effects in mouse and human CD4^+^ T cells have been reported. In mice, Sema4A is involved in T cell priming and Th1 skewing, both processes mediated by Tim-2 [14,20]. Besides that, Sema4A deficiency induces Th2 skewing and enhances the severity of allergic responses [21,22,23]. Finally, Sema4A induced the expression of IL-17 by CD4^+^ T cells from autoimmune encephalomyelitis mice [11,24]. In human CD4^+^ T cells, exogenous Sema4A induces ILT-4 mediated Th2 differentiation and, on the contrary to mice studies, inhibits Th1 skewing. Moreover, we also found that Sema4A secreted by activated CD4^+^ T cells of SSc patients is able to induce the expression of Th17 cytokines in a PlexinB2, PlexinD1 and NRP-1 dependent manner [12,13].

Altogether, these data demonstrate that exogenous Sema4A is involved in human naïve Th cell differentiation. However, whether autocrine Sema4A is involved on this process and the functional consequences of its inhibition are still unknown.

## 2. Results

### 2.1. Sema4A Is Induced During CD4^+^ T Cell Differentiation

Although previous reports have shown that Sema4A is not expressed in naïve CD4^+^ T and only induced after CD4^+^ T cells activation [12,13], its expression during Th cell differentiation is still unknown. Indeed, unstimulated naïve CD4^+^ T cells did not express SEMA4A. However, upon Th1, Th2 and Th17 differentiation SEMA4A expression is upregulated at day 3, although differences were only significant in the Th2 subset. Despite the lower gene expression levels compared to day 3, at day 7 we still observed an upregulation of SEMA4A (Figure 1A). In addition, we also determined the effect of Th differentiation on the secretion of Sema4A. At day 3, the Sema4A levels were already elevated in the three Th cell subsets compared to the non-stimulated naïve CD4^+^ T cells, even though it was only significant in the Th17 subset. At day 7, the secretion of Sema4A was significantly higher in the three differentiated Th subsets compared to the non-differentiated naïve T cells (Figure 1B).

### 2.2. Autocrine Sema4A–plexinD1 Signaling Induces Th2 and Th17 Differentiation and Suppresses Th1 Skewing

We have previously shown that neutralization of autocrine/paracrine Sema4A signaling reduced the expression of Th17 cytokines by the total CD4^+^ T cells [13], thus we next analyzed the effect of the inhibition of this signaling during Th helper cell differentiation. For this purpose we used an anti-PlexinD1 neutralizing antibody, as PlexinD1 is the most expressed Sema4A receptor in naïve CD4^+^ T cells [13]. Compared to the respective isotype control, the neutralization of PlexinD1 induced the expression of IFNG in Th1-diferentiated cells, while it reduced the expression of Th2 cytokines (IL4 and IL5), and Th17 cytokines (IL17 and IL22) cytokines by Th2 and Th17 differentiated cells, respectively. TNF expression was upregulated in the three cell subsets but the expression was not modulated by the PlexinD1 neutralization (Figure 2). Consistently, we also found these effects earlier in the differentiation process, as at day 3 PlexinD1 blocking upregulated IFNG levels and reduced IL4 and IL17 expression in the respective Th subsets, Th1, Th2 and Th17 (Appendix A).

We next confirmed these results at the protein level; on Th1 cells, PlexinD1 neutralization significantly increased the secretion of IFNγ and showed a trend towards a higher production of this cytokine after restimulation with PMA/ionomycin. On the other hand, neutralization of the Sema4A-PlexinD1 signaling significantly reduced the secretion and the production of IL-4 and IL-17 by Th2 and Th17 cells, respectively (Figure 3A,B and Appendix A). We also observed a reduction on IL-22 production by Th17 cells, but differences were not significant (Figure 3B).

To rule out the possibility of unspecific effects due to cell viability and/or proliferation, we analyzed the viability of a naïve CD4^+^ T cell and proliferation during differentiation. At day 3, non-differentiated naïve T cells did not proliferate, but in the Th1, Th2 and Th17 subsets, the proliferation rate was nearly 100%, independent of PlexinD1 neutralization (Appendix A). Overall, cell viability was similar through all the culture conditions either at day 3 and day 7, without effect of the PlexinD1 neutralization. Nevertheless, the overall cell viability at day 7 was slightly lower compared with day 3 (Appendix A).

### 2.3. PlexinD1 Neutralization Regulates TBX21, GATA3 and RORC Expression

We next analyzed whether PlexinD1 neutralization also modulates the expression of T-bet, GATA3 and RORγt, the master transcription factors involved in Th cell differentiation [3]. The expression levels of TBX21, the gene encoding T-bet, was increased on Th1-differentiated cells at day 3 and 7 of differentiation in the presence of the anti-PlexinD1 antibody. On the contrary, GATA3 expression was reduced on Th2-differentiated cells at both day 3 and 7, although differences were not significant. Similarly, PlexinD1 neutralization significantly reduced the expression of RORC, the gene that encodes RORγt, in Th17 cells at day 3 and 7 (Figure 4A,B). Therefore, our data suggest that Sema4A-PlexinD1 axis impacts the expression of master transcription factors involved in the Th cell differentiation processes.

### 2.4. Sema4A-PlexinD1 Blockade Reduces PlexinB2 Expression during Th17 Cell Differentiation

As Sema4A has multiple receptors [10], we also determined whether blockade of the Sema4A-PlexinD1 axis modulates the expression of Sem4A receptors. The neutralization of PlexinD1 did not affect the mRNA expression of PlexinD1, PlexinB1, NRP-1 or ILT-4 in any of the subsets analyzed at day 7 (Figure 5A). However, PlexinD1 blockage led to a significant reduction in PLEXINB2 expression in Th17-diferentiated cells at both day 3 and 7 (Figure 5B,C) and to a lower percentage of PlexinB2 positive cells at day 3 of Th17 differentiation (Figure 5D and Appendix A).

### 2.5. PlexinD1 Neutralization Regulates CD4^+^ T Cells Cytokine Production Induced by Systemic Sclerosis Patient Serum

Previously we have shown that Sema4A is elevated in the serum of SSc patients and it induces the production of Th17 cytokines [13]. In addition, both IL-17 and IL-4 cytokines are elevated in SSc patients and have been shown to play an important role in SSc pathogenesis [25,26,27,28,29]. Thus, in order to confirm the functional role of the Sema4A-PlexinD1 axis on a CD4^+^ T cell-mediated disease, we investigated the role of PlexinD1 neutralization on the CD4^+^ T cells cytokine production induced by the serum of SSc patients. In CD4^+^ T cells activated in the presence of SSc-serum, the neutralization of PlexinD1 significantly induced the secretion of IFN-γ, while the secretion of IL-17 was significantly reduced (Figure 6). The secretion of IL-4 despite the low production and undetectable in some donors, shows a trend towards a reduction after the neutralization of PlexinD1 (Figure 6). Altogether, these data suggest that Sema4A-PlexinD1 signaling is involved in the elevated levels of IL-17 and IL-4 observed in these patients.

## 3. Discussion

Here, we demonstrated the role of autocrine Sema4A-PlexinD1 axis on CD4^+^ T cell differentiation and its contribution to T cell-mediated diseases. Previous reports have shown that Sema4A is expressed by activated CD4^+^ T cells and memory Th2 cells [12,13]. Here we first show that the expression of Sema4A was induced during the CD4^+^ T differentiation and, together with the fact that Sema4A was involved in T cell activation and differentiation, suggest that autocrine Sema4A may be involved in these processes.

Secondly, we demonstrated that the autocrine Sema4A-PlexinD1 axis was a negative regulator of human Th1 differentiation and played a crucial role in Th2 skewing. These findings are in accordance with the work of Lu N et al. [12], in which exogenous Sema4A administration repressed Th1 differentiation and induced Th2 skewing. In addition, we unprecedently showed that Sema4A-PlexinD1 axis is also involved on Th17 differentiation, which is in line with previous works, which demonstrate that Sema4A induces IL-17 production in both mouse and human CD4^+^ T cells [11,13,24]. Importantly, the effect of the Sema4A-PlexinD1 axis on T helper cell differentiation is supported by the regulation of the master transcription factors T-bet, GATA3 and RORC. Similarly to our previous work [13], but on the contrary to the work of Lu N et al. [12], the autocrine Sema4A-PlexinD1 signaling did not affect the CD4^+^ T cell proliferation. These differences may be attributed to the stimulation with exogenous Sema4A, the use of CD4^+^ T cells from buffy coats and the different manner of CD4^+^ T cell activation, as these authors used an anti-CD3 antibody at a suboptimal concentration and did not activated the CD28 signaling pathway. Altogether, these data suggest that the Sema4A-PlexinD1 axis might be implicated in the maintenance of the homeostatic levels of Th1, Th2 and Th17 cells and therefore in the proper host defense against intracellular and extracellular pathogens [1,2]. This role of Sema4A-PlexinD1 was observed in combination with key cytokines involved on Th differentiation, such as IL12 (Th1), IL-4 (Th2) and IL-23 (Th17) [4,5,9]. However, whether Sema4A is capable of modulating Th differentiation in the absence of these cytokines or is a costimulatory pathway implicated in Th skewing needs to be further elucidated.

Sema4A binds to different receptors in a cell type and context dependent manner [10,30]. Here we found that PlexinD1 is involved in the differentiation of three CD4^+^ T subsets. To our knowledge PlexinD1 is the only Sema4A receptor involved in the inhibition of Th1 differentiation, while in the Th2 skewing both PlexinD1 and ILT-4 are implicated [12]. The neutralization of Sema4A-PlexinD1 axis downregulated the expression of PlexinB2 in Th17 cells, and as PlexinB2 is involved in the secretion of Th17 cytokines by activated CD4^+^ T cells [13], the Sema4A-PlexinB2 axis seems also to be implicated in the Th17 skewing. Hence, Sema4A plays a central role in CD4^+^ T cell differentiation through the involvement of different receptors. Nevertheless, the role of other Sema4A receptors involved in T cell function, such as NRP-1, PlexinB1 or PlexinB2 needs to be elucidated.

Lastly, we translated our findings to a CD4^+^ T cell-mediated disease and we demonstrated that the neutralization of PlexinD1 suppressed the CD4^+^ T cell secretion of IL-17 and IL-4 induced by the serum of SSc patients. Although PlexinD1 can bind to other semaphorin family members, mainly Sema3E [10,31], but also Sema3C, Sema3D and Sema3G [32,33,34], Sema4A is the only known PlexinD1 ligand able to induce T cell activation. This fact, together with the elevated Sema4A expression observed in SSc patients [13], suggest that the Sema4A-PlexinD1 axis is responsible, at least in part, for the elevated IL-4 and IL-17 levels observed in these patients. As Sema4A expression is also elevated in other CD4^+^ T cell-mediated diseases including, asthma, RA and MS [11,12,19,24], it is tempting to speculate that Sema4A-PlexinD1 signaling is involved in the dysregulation of Th cell homeostasis observed in these diseases and targeting this axis might be a beneficial therapeutic approach. As binding of PlexinD1 to its different ligands is not only involved in T cell differentiation, but also plays a key role in vascular, cardiac, skeletal and neuronal development and homeostasis [35,36,37,38,39,40], we consider that neutralizing Sema4A might be a better therapeutic option than blocking PlexinD1. A limitation of our study is that we could not test this possibility, due to the lack of commercially available anti-Sema4A neutralizing antibodies. Therefore, further studies using other methodologies such as gene silencing or deletion are needed to elucidate the pathological role of Sema4A on CD4^+^ T cell-mediated diseases.

## 4. Materials and Methods

### 4.1. Naïve CD4^+^ T Cell Isolation

Peripheral blood mononuclear cells (PBMCs) from healthy controls were isolated by the Ficoll gradient (GE Healthcare, Zwijndrecht, The Netherlands). Cells were further isolated using the CD4+ T Cell Isolation Kit on an autoMACS Pro Separator, according to the manufacturer’s instructions (Miltenyi Biotec, Leiden, The Netherlands). To isolate the naïve subpopulation, total CD4^+^ T cells were stained with anti-CD4 APC-eF780 (eBioscience, Nieuwegein, The Netherlands), anti-CD27 BV510 (Biolegend, Amsterdam, The Netherlands), anti-CD25 PE, anti-CD127 Alexa Fluor-647 and anti-CD45RO PE/Cy7 (BD Biosciences, Vianen, The Netherlands) antibodies and purified on a BD FACSAria™ III cell sorter (BD Biosciences, Vianen, The Netherlands). Naïve CD4 ^+^ CD25^−^CD27 ^+^ CD45RO^−^ T cells were defined according to the gating strategy shown in Appendix A. Purity was consistently > 99%.

### 4.2. Naïve CD4^+^ T Cell Differentiation

Naïve CD4^+^ T cells were preincubated for 1 h with the neutralizing antibody anti-PlexinD1 or its respective isotype control, goat IgG (both 2.5 μg/mL, R&D systems, Abingdon, United Kingdom) in RPMI-GlutaMAX (Thermo Fisher Scientific, Nieuwegein, The Netherlands) supplemented with 10% fetal bovine serum (FBS, Biowest, Amsterdam, The Netherlands), 10.000 I.E. penicillin-streptomycin (Thermo Fisher Scientific, Nieuwegein, The Netherlands) and 1:500 Primocin (Invivogen, Toulouse France). Next, cells were activated with Dynabeads Human T-Activator CD3/CD28 (Thermo Fisher Scientific, Nieuwegein, The Netherlands) at a bead-to-cell ratio of 1:5 and differentiated either to Th1, Th2 or Th17 for 7 days with the specific T helper cell differentiation cocktails (Appendix A). At day 3, half of the medium was replaced with fresh medium containing the differentiation cocktails and the anti-PlexinD1 antibody or the isotype control. For the intracellular cytokine staining, 1 µg/mL of phorbol 12-myristate 13-acetate (PMA), 50 ng/mL of ionomycin (both from Sigma Aldrich, Zwijndrecht, The Netherlands) and 1 µg/mL of GolgiStop (BD Biosciences, Vianen, The Netherlands) were added for the final 4 h of stimulation.

### 4.3. CD4^+^ T Cell Stimulation

Total CD4^+^ T cells were pretreated for 1 h with the anti-PlexinD1 antibody or its respective isotype control IgG (both 2.5 μg/mL) and then activated with Dynabeads Human T-Activator CD3/CD28 (Thermo Fisher Scientific, Nieuwegein, The Netherlands) at a bead-to-cell ratio of 1:5 and incubated with the serum of SSc patients (20% *v/v*), for 5 days. All included patients fulfilled the ACR/EULAR 2013 classification criteria for SSc [41]. All patients provided informed written consent approved by the local institutional medical ethics review boards prior to inclusion in this study (NL47151.041.13, on 29 November 2013). Samples and clinical information were treated anonymously immediately after collection. Demographics and clinical characteristics of the included patients are detailed in Appendix A.

### 4.4. Flow Cytometry

CD4^+^ T cells were stained with Fixable Viability Dye (eBioscience, Nieuwegein, The Netherlands), and antibodies for PlexinB2 APC and its respective isotype control (both from R&D systems, Abingdon, United Kingdom). Alternatively, cells were fixed and permeabilized using the Foxp3/Transcription Factor Staining Buffer Set (eBioscience, Nieuwegein, The Netherlands), and stained for IL-17A FITC, IL-22 APC, IFNγ PerCP-Cy5.5 (all from eBioscience, Nieuwegein, The Netherlands), IL-4 BV711, IL-13 PE and TNF BV421 (all from BD Biosciences, Vianen, The Netherlands). For proliferation analysis, CD4^+^ T cells were labeled with CellTrace Violet (1.5 µM, Thermo Fisher Scientific, Nieuwegein, The Netherlands) prior to culture. Samples were acquired on a BD LSR Fortessa (BD Biosciences), or on a BD FACSCanto (BD Biosciences, Vianen, The Netherlands) using the BD FACSDiva software (BD Biosciences, Vianen, The Netherlands). FlowJo software (BD Biosciences, Vianen, The Netherlands) was used for data analyses. All flow cytometry data is presented as the percentage of positive cells.

### 4.5. Cytokine Measurement

Sema4A (Biomatik LLC, Wilmington, DE, USA), IL-17A, IFN-γ (eBioscience, Nieuwegein, The Netherlands) and IL-4 (R&D systems, Abingdon, United Kingdom) were measured by ELISA in cell-free supernatants, according to the manufacturer’s instructions.

### 4.6. RT-PCR and Quantitative qPCR

RNA was isolated using the RNeasy micro Kit and RNase-Free DNase Set (Qiagen, Venlo, The Netherlands). Total RNA was reverse-transcribed using an iScript cDNA Synthesis kit (Biorad, Veenendaal, The Netherlands). qPCR reactions were performed using SYBR green (Applied Biosystem, Nieuwerkerk aan Den Ijssel, The Netherlands) with a StepOne Plus Real-Time PCR system (Applied Biosystems, Nieuwerkerk aan Den Ijssel, The Netherlands). cDNA was amplified using specific primers (all from Integrated DNA Technologies, Inc. (IDT), Leuven, Belgium, see Appendix A). Relative levels of gene expression were normalized to the *B2M* housekeeping gene. The relative quantity of mRNA was calculated using the formula 2^−Δ*C*t^ × 1000.

### 4.7. Statistical Analysis

Statistical analysis was performed using GraphPad Prism 8 (GraphPad Software, Inc., San Diego, CA, USA). Potential differences between experimental groups were analyzed by a parametric ANOVA test. *p* < 0.05 was considered statistically significant.

## 5. Conclusions

The Sema4A-PlexinD1 axis is an important regulator of T helper cell differentiation and therefore regulating Sema4A levels might be a potential therapeutic approach in CD4^+^ T cell-mediated diseases.

## Figures and Tables

**Figure 1 ijms-21-06965-f001:**
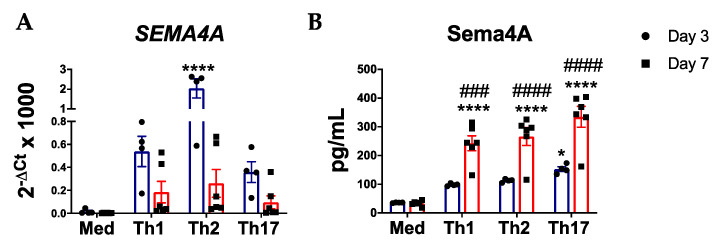
Sema4A is induced during T helper cell differentiation. Sema4A mRNA (**A**) and protein (**B**) levels in unstimulated naïve CD4^+^ T cells or in Th1, Th2 or Th17 differentiated cells for 3 days (*n* = 4) and 7 days (*n* = 6). Means and SEM are shown. * *p* < 0.05 and **** *p* < 0.0001 compared to the medium at day 3. ### *p* < 0.001 and #### *p* < 0.0001 compared to the medium at day 7.

**Figure 2 ijms-21-06965-f002:**
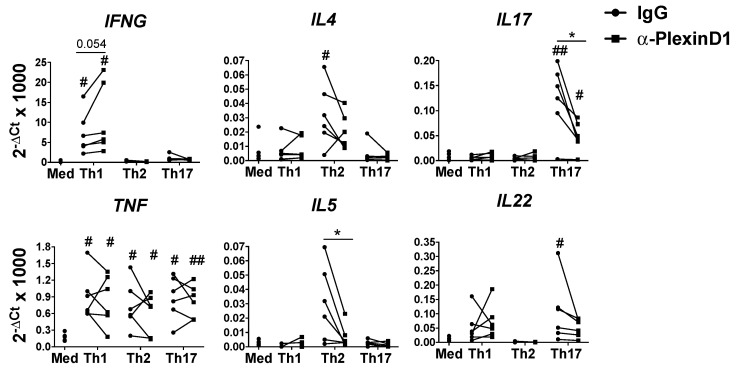
Sema4A-PlexinD1 axis inhibits Th1 and promotes Th2 and Th17 differentiation. Cytokines mRNA expression in naïve CD4^+^ T cells differentiated either in medium (Med), Th1, Th2 or Th17 differentiation cocktails in the presence of an anti-PlexinD1 antibody or its respective isotype control for 7 days (*n* = 6). Data is presented as connected dots. * *p* < 0.05. # *p* < 0.05 and ## *p* < 0.01 compared to the medium.

**Figure 3 ijms-21-06965-f003:**
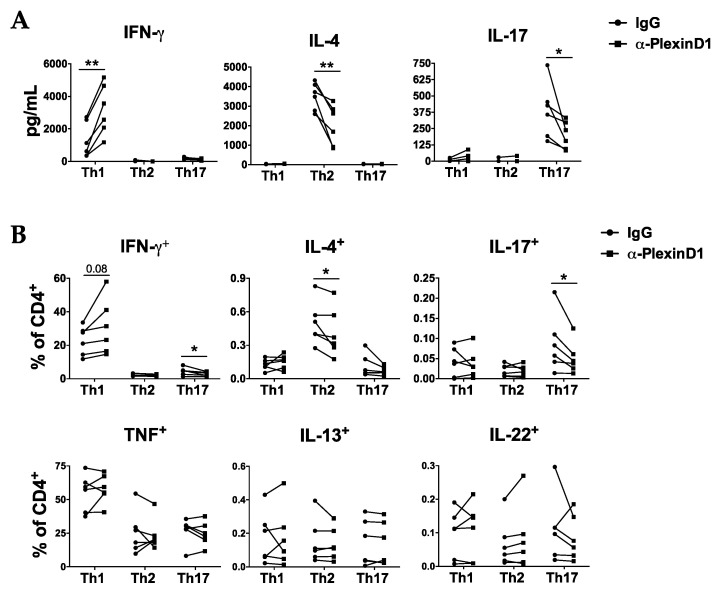
Sema4A-PlexinD1 axis inhibits Th1 and promotes Th2 and Th17 differentiation. Cytokine secretion (**A**) and intracellular cytokine production (**B**) in naïve CD4^+^ T cells differentiated with Th1, Th2 or Th17 differentiation cocktails in the presence of an anti-PlexinD1 antibody or its respective isotype control for 7 days (*n* = 6). Data is presented as connected dots. * *p* < 0.05 and ** *p* < 0.01.

**Figure 4 ijms-21-06965-f004:**
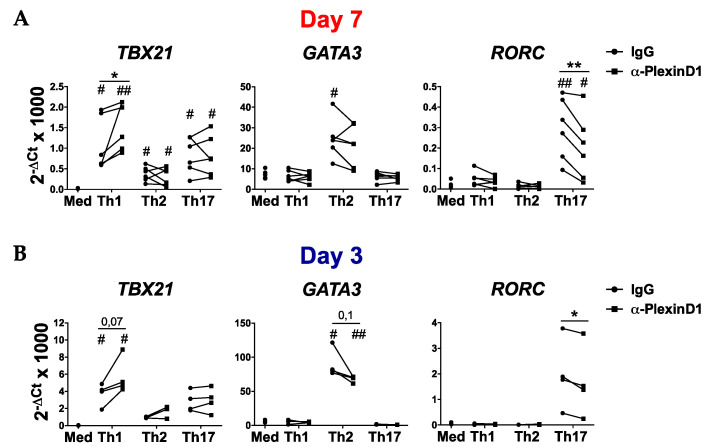
Sema4A-PlexinD1 axis regulates the expression of transcription factors involved in T helper cell differentiation. (**A,B**) TBX21, GATA3 and RORC expression in naïve CD4^+^ T cells differentiated either in medium or Th1, Th2 or Th17 differentiation cocktails in the presence of an anti-PlexinD1 antibody or its respective isotype control for 7 days (**A**, *n* = 6) and 3 days (**B**, *n* = 4). Data is presented as connected dots. * *p* < 0.05 and ** *p* < 0.01. # *p* < 0.05 and ## *p* < 0.01 compared to the medium.

**Figure 5 ijms-21-06965-f005:**
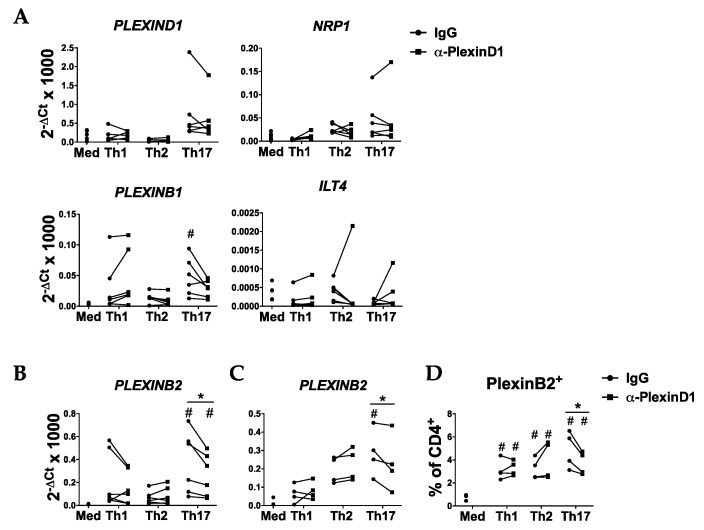
Sema4A-PlexinD1 axis induces the expression of PlexinB2 in Th17 cells. (**A**) mRNA expression of Sema4A receptors in naïve CD4^+^ T cells differentiated in medium or Th1, Th2 or Th17 differentiation cocktails in the presence of an anti-PlexinD1 antibody or its respective isotype control for 7 days (*n* = 6). (**B**–**D**) PlexinB2 mRNA (**B,C**) and protein (**D**) expression in naive CD4^+^ T cells differentiated in medium or Th1, Th2 or Th17 differentiation cocktails in the presence of an anti-PlexinD1 antibody or its respective isotype control for 7 days (**B**, *n* = 6) and 3 days (**C,D**, *n* = 4). Data are presented as connected dots. * *p* < 0.05. # *p* < 0.05 compared to the medium.

**Figure 6 ijms-21-06965-f006:**
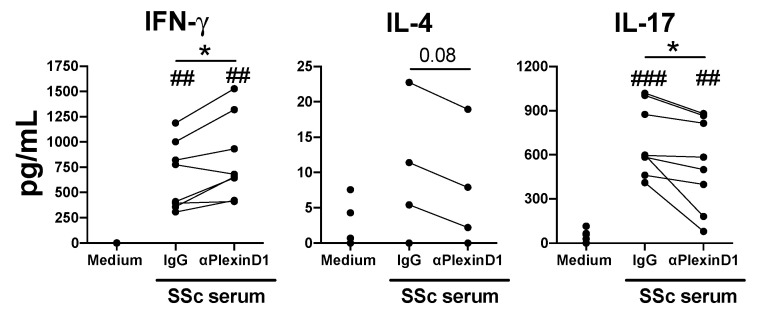
Sema4A-PlexinD1 axis regulates CD4^+^ T cytokine production induced by systemic sclerosis patient serum. IFN-γ, IL-4 and IL-17 secretion by activated CD4^+^ T cells, pretreated for 1 h with an anti-PlexinD1 antibody or its respective isotype control (IgG) and incubated with the serum of SSc patients (20% *v/v*) for 5 days (*n* = 8). Data are presented as connected dots. * *p* < 0.05. ## *p* < 0.01 and ### *p* < 0.001 compared to medium.

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
