# Peer review of "Semaphorin4A-Plexin D1 Axis Induces Th2 and Th17 While Represses Th1 Skewing in an Autocrine Manner"

_ijms, 2020, doi:10.3390/ijms21186965_

Round 1

Reviewer 1 Report

In this manuscript authors divulge the implication of Sema4A and PlexinD1 on CD4 T cell differentiation and regulation of Sema4A4-PlexinD1 axis contribution to T cell-mediated diseases. Over all the manuscript is well written and all the experiments and results are discussed well. I would ask for some minor revision before acceptance in principle to simplify the graphs in result section for the scientific readers. Many of them in Fig-2 and 3.

Author Response

In this manuscript authors divulge the implication of Sema4A and PlexinD1 on CD4 T cell differentiation and regulation of Sema4A4-PlexinD1 axis contribution to T cell-mediated diseases. Over all the manuscript is well written and all the experiments and results are discussed well. I would ask for some minor revision before acceptance in principle to simplify the graphs in result section for the scientific readers. Many of them in Fig-2 and 3.

We thank the reviewer for his/her positive statements about our work. To simplify Figures 2 and 3 we have moved some of the panels to supplementary methods. We hope that the figures are now clearer and easier to understand.

Reviewer 2 Report

The manuscript is interesting and well written.  However, I suggest to discuss and add as reference the paper by Murdaca et al. upon Th17 in chronic immune-mediated diseases. Furthermore, what is the link between the semaphorin and free radicals which play a role in systemic sclerosis and other autoimmune diseases associated accelerated atherosclerosis (see and add as reference paper by Murdaca  et al concerning free radicals and accelerated atherosclerosis)

Author Response

The manuscript is interesting and well written.  However, I suggest to discuss and add as reference the paper by Murdaca et al. upon Th17 in chronic immune-mediated diseases. Furthermore, what is the link between the semaphorin and free radicals which play a role in systemic sclerosis and other autoimmune diseases associated accelerated atherosclerosis (see and add as reference paper by Murdaca  et al concerning free radicals and accelerated atherosclerosis).

 We thank the reviewer for his/her positive statements about our work. We have discussed and added the suggested reference in the new version of the review. Regarding the link between Sema4A and free radicals is a very interesting topic and, in fact, other semaphorins (Sema3F, Sema4D among others) are implicated in the production of free radicals. However this manuscript is focused in the role of Sema4A-PlexinD1 axis on Th differentiation and the production of Th1/Th2/Th17 cytokines. For that reason we believe that discussing the potential link between Sema4A and free radicals is out of the scope of this work. In fact, for that reason we did not discuss the role of Sema4A in other processes involved in SSc pathogenesis, such as angiogenesis and cell migration and invasion.